# Relationship between Solitary Masturbation and Sexual Satisfaction: A Systematic Review

**DOI:** 10.3390/healthcare12020235

**Published:** 2024-01-17

**Authors:** Oscar Cervilla, Ana Álvarez-Muelas, Juan Carlos Sierra

**Affiliations:** Mind, Brain, and Behavior Research Center (CIMCYC), University of Granada, 18011 Granada, Spain; ocervilla@ugr.es (O.C.); alvarezm@ugr.es (A.Á.-M.)

**Keywords:** solitary masturbation, sexual satisfaction, sexual health, sexual functioning, sexual relationships, systematic review

## Abstract

Masturbation is a healthy sexual behavior associated with different sexual functioning dimensions, which highlights sexual satisfaction as an important manifestation of sexual wellbeing. This review aims to systematically examine studies that have associated masturbation with sexual satisfaction, both in individuals with and without a partner. Following the PRISMA statement, searches were made in the APA PsycInfo, Medline, Scopus, and Web of Science databases. The search yielded 851 records, and twenty-two articles that examined the relation between solitary masturbation and sexual satisfaction were selected. In men, a negative relation between masturbation and sexual satisfaction was observed in 71.4% of the studies, 21.4% found no such relation, and 7.2% observed a positive association. In women, 40% reported no relation, 33.3% a negative relation, and 26.7% a positive one. The negative association between solitary masturbation and sexual satisfaction is consistent with the previously proposed compensatory role of masturbation, especially for men. In women, compared to men, the complementary role of masturbation in relation to sexual relationships is observed to a greater extent and is associated more closely with sexual health. The importance of including different parameters beyond the masturbation frequency in future studies to explore its relation with sexual satisfaction is emphasized. This systematic review is registered in PROSPERO (CRD42023416688).

## 1. Introduction

Masturbation is a healthy sexual behavior practiced with others (e.g., a partner) or individually [1]. Solitary masturbation is defined as erotic self-stimulation without anyone else being present or participating [2]. Its practice is present from very early development phases to old age [3]. This behavior favors self-exploration and sexual learning in a context in which the presence of sexual difficulties might be less prevalent [4]. Previous studies have stressed the importance of solitary masturbation for the adjustment and generalization of the sexual response to the context of sexual relationships [5], acting as a therapeutic tool to deal with some sexual difficulties [6,7].

The relation of solitary masturbation with sexual relationships has been studied mostly by two models: compensatory and complementary. The compensatory model hypothesizes that masturbation frequency could increase for the purpose of substituting unsatisfactory or insufficient sexual relationships [8,9]. The complementary model considers a positive relation between masturbation behavior and sexual relationships, implying that practicing one would be associated with the other one being practiced more frequently [9]. Previous pieces of evidence suggest that the compensatory pattern would be more present in men, with the complementary pattern in women [9,10,11,12,13,14], despite some studies showing the independence of gender in both of these models [15,16].

Masturbation has been related to different sexual functioning dimensions, although very few results have been obtained. Positive associations have been described with sexual desire [17], sexual arousal [11], or orgasm [5], which evidences the positive implication of this behavior in sexual response. One of the most interesting dimensions is sexual satisfaction, which is an important indicator of sexual health [18,19,20].

Sexual satisfaction could be considered the last phase of the sexual response cycle according to Basson’s model [21,22] and is defined as “an affective response arising from one’s subjective evaluation of the positive and negative dimensions associated with one’s sexual relationship” [23] (p. 268). Its study requires a multidimensional approach that contemplates personal, interpersonal, and social factors [19,24]. In line with this, the Ecological Theory of Human Development [25] has served as a guide to study it by bearing in mind the different associated relevant variables, which range from the closest to the most distant to an individual [19]. Of the variables associated with sexual satisfaction, solitary masturbation falls under personal-type factors [13,26,27].

As far as we are aware, the pieces of evidence that have associated solitary masturbation with sexual satisfaction have not been integrated, despite its importance for sexual health. Thus, considering that previous literature reviews on this are missing, the objective of the present study is to systematically analyze the results obtained in the scientific literature about the relation between solitary masturbation (i.e., its presence/absence and/or frequency) and sexual satisfaction, including a comparison of this relation in men and women.

## 2. Materials and Methods

This systematic review was conducted in accordance with the Preferred Reporting Items for Systematic Reviews and Meta-Analyses Guidelines (PRISMA) [28]. The protocol of this review is registered in PROSPERO (registration number: CRD42023416688).

### 2.1. Eligibility Criteria

To fulfill the objectives of this systematic review, the considered studies had to address the relation between solitary masturbation and sexual satisfaction. Eligible studies had to meet all the following inclusion criteria: (a) original research articles; (b) solitary masturbation (as presence/absence or frequency); (c) sexual satisfaction was assessed using standardized instruments, ad hoc items, or derived from scales, questionnaires, or interviews; (d) they had examined the direct and indirect relation, considering mediators and/or covariates between solitary masturbation and sexual satisfaction.

There was no limitation for publication year, and the English and Spanish languages were considered.

### 2.2. Information Sources

The literature search was conducted on APA PsycInfo, Medline, Scopus, and Web of Science. The last database query date was 30 October 2023.

### 2.3. Search Strategy

Following the recommendations by Quevedo-Blasco [29] and using the terms related to sexual satisfaction as employed in the systematic review by Sánchez-Fuentes et al. [19], the search strategy integrated the following terms: (masturb* OR self-stimulat* OR onanism* OR “solitary sexual activit*”) AND (“satisfac* sex*” OR “sex* satisfact*” OR “satisfaction with sex*”), using the truncation “*” to include any variant of words.

To validate the search strategy, a peer review was conducted by proofreading the syntax, spelling, and structure and ensuring that the search formula identified articles that were relevant to the search. The formula was applied to the title, abstract, and/or keywords, or, if applicable, to the topic, to narrow down the search on the topic of masturbation and sexual satisfaction.

### 2.4. Selection Process

The search results were exported on the Rayyan online platform, a web-based automated screening tool developed by the Qatar Computing Research Institute (QCRI) that is accessible at www.rayyan.ai (accessed on 15 November 2023) [30]. This export included the title, authors, publication date, abstract, and keywords. Two authors (AÁM and OC) independently reviewed the documents based on their title, abstract, and keywords by categorizing articles as included, doubtful, or rejected. The studies labeled as doubtful underwent a full-text review, and discrepancies were solved by consensus. Final decisions, if necessary, were made by a third researcher (JCS).

### 2.5. Data Collection Process

The articles that met the inclusion criteria were comprehensively read independently by two reviewers to guarantee the objectivity and rigor of the results. A data collection form was designed, and the extracted data were compared to any discrepancies resolved by discussion. The extracted data included: (a) authors, (b) country, (c) sample, (d) participants’ sexual orientation, (e) instrument used to assess solitary masturbation, (f) instrument applied to assess sexual satisfaction, and (g) results about the association between masturbation and sexual satisfaction. The true Kappa value was employed to assess the reliability of coding [31,32]. Intercoding was evaluated by indicating agreement or disagreement in the analyses of the categories extracted during the article selection process [33]. A true Kappa value of 0.91 was obtained when considering the agreement between coders to be satisfactory with a Kappa value above 75%.

### 2.6. Data Items

Outcome measures that assess (a) solitary masturbation and (b) sexual satisfaction were extracted. The results can be reported as the presence/absence of solitary masturbation by dichotomous items, a frequency scale of solitary masturbation, or interviews. Likewise, an overall test score to provide a general measure of sexual satisfaction (e.g., general sexual satisfaction) or subscales/specific items to provide a measure of domain-specific sexual satisfaction (i.e., physical sexual satisfaction) was/were considered.

### 2.7. Study Risk of Bias Assessment

The risk of bias in the included studies was assessed using the Quality Assessment Tool for Observational Cohort and Cross-Sectional Studies (QATOCCS) [34] for those studies that indicated a quantitative methodology and the Strengthening the Reporting of Observational Studies in Epidemiology (STROBE) [35] tool for the studies that indicated an observational analytical methodology. These tools provided a standardized framework for assessing the scientific rigor of all the reviewed studies through a checklist of requirements (e.g., definition of the study population, the research question, control definition, inclusion criteria, blindness, and the reporting of confounders). The evaluation ensured the studies’ robustness and the results’ reliability. To do so, two authors independently applied the tools to the included studies. If discrepancies arose, they were solved by consensus.

### 2.8. Synthesis Methods

Table 1 shows the individual results of the studies and the synthesis. For better visualization purposes, the authors, publication year, country, sample size, assessment of masturbation and sexual satisfaction, and the main findings about the relation between both variables were tabulated.

## 3. Results

### 3.1. Study Selection

The database search yielded 851 records (see Appendix A). After eliminating duplicates, 464 records remained according to their title, abstract, and keywords. Of these, 432 records were excluded due to the exclusion criteria. A total of 32 underwent a full-text examination, and, finally, 10 were eliminated because they did not meet the inclusion criteria. To broaden the selection process, although a search was made for the papers cited in the studies to be considered, none of them were included. This left 22 papers that met the inclusion criteria and methodological quality standards and could, therefore, be included in the present systematic review. Figure 1 shows a flow chart of the selection process for these studies.

Below are the results of the 22 analyzed papers that evaluated the relation between solitary masturbation and sexual satisfaction (see Table 1).

### 3.2. Sociodemographic Characteristics

The studies were conducted in one or more of these countries: the United States (5 publications), Norway (4), Brazil (2), Switzerland (1), Sweden (1), Denmark (1), Belgium (1), Portugal (1), Hungary (1), the Czech Republic (1), the United Kingdom (1), Australia (1), Malaysia (1), China (1), Canada (1), and Germany (1).

Seventeen of the twenty-two papers included both men and women samples [12,15,16,37,41,46,47,48,49,55,57,59,62,64,65,66,68], while four papers were conducted exclusively with women [36,40,42,44] and one with men [53]. Three studies reported exclusively heterosexual participants [47,48,53], and six also included populations of other sexual orientations (e.g., gay or bisexual) [16,41,42,46,55,66]. The rest of the studies did not report their participants’ sexual orientation.

### 3.3. Instruments to Assess Masturbation

Most of the studies used ad hoc procedures to assess masturbation: frequency scales and, to a lesser extent, a dichotomous item or an interview to determine presence/absence. Only three papers employed an item drawn from validated scales or found in previous projects to assess masturbation frequency [12,46,47]. The time frame to which masturbation practice referred, in those studies that indicated it, was variable: in the last 24 h [66], in the last month [12,36,37,46,59,65], in the last 6 months [41,49,53], or in the last year [15,42].

Regarding the response scale, except for two studies in which presence/absence was evaluated dichotomously (i.e., having masturbated vs. not having masturbated) [15,66] and one in which the response was free (i.e., indicate the number of times) [36], in the remaining papers that specified it, Likert-type response scales of three [59], four [55], five [48], six [16,41,49,68], seven [12,46,57], eight [53,62], nine [42,44], and ten [64] categories were used.

### 3.4. Instruments to Assess Sexual Satisfaction

Sexual satisfaction was assessed in twelve of the studies using ad hoc items on satisfaction with sexual relationships and/or sex life [12,15,16,36,40,41,46,47,48,49,64,68], answered with a Likert-type scale, except for two studies that employed dichotomous items (i.e., satisfied vs. not satisfied) [15,40].

Four papers employed items drawn from one of the following validated instruments or more: the Life Satisfaction Scale [38,39], the Multidimensional Sexuality Questionnaire (MSQ) [45], the Female Sexual Function Inventory (FSFI) [63], the Changes in Sexual Functioning Questionnaire (CSFQ) [61], the Satisfaction with Sex Life Scale—Revised [58], and the Derogatis Interview for Sexual Functioning (DISF-SR) [60].

The remaining six papers used standardized assessment instruments: the Global Measure of Sexual Satisfaction [54] the Female Sexual Quotient [43], which were both included in two papers, the Male Sexual Quotient [56], and the Multidimensional Sexuality Questionnaire (MSQ) [45].

### 3.5. Relation between Masturbation and Sexual Satisfaction

Five studies (22.7%) examined the relation between masturbation and sexual satisfaction in men and women together. They revealed a negative relation [55,57,62,68] or no relation [57,66] between both variables.

Of the studies with samples exclusively made up of men or that examined men independently of women, 71.4% of them (ten articles) reported a negative relation between masturbation and sexual satisfaction [12,16,37,41,44,47,49,53,59,64]. In contrast, three studies (21.4%) found no significant relation between the two variables [46,48,65], and a single study (7.2%) observed a positive association between masturbation and sexual satisfaction [15].

Of the studies with samples formed exclusively of women or that examined women independently of men, six (40%) indicated no relation between masturbation and sexual satisfaction [40,41,46,48,59,65], five studies (33.3%) reported a negative relation [12,16,36,37,44], and four (26.7%) showed a positive relation [15,42,46,49].

## 4. Discussion

Solitary masturbation is a behavior with implications for sexual health, among which sexual satisfaction is included. To integrate the results obtained from the scientific literature about the relation between solitary masturbation and sexual satisfaction, this study presents a systematic review of the articles published up to October 2023. Most of the studies included in the review (63.6%) were conducted in the United States and Europe. This aligns with the increasingly positive view in western countries that solitary masturbation is considered to be a source of pleasure that is independent of sexual relationships [1,3,10,69]. The evolution toward a positive view of this behavior in recent years has promoted further research, which is reflected by the publication year of the works included in this systematic review because most had publication dates in the last two decades. Nevertheless, masturbation experiences can be positive or negative, depending on prevailing social attitudes [1]. The cultural divide observed in this review could be evidence of the challenges in the area of research into sexuality that some societies face, such as African ones, where difficulties are reported for people to share some aspects related to their sexuality [70]. Masturbation is still taboo in some of these societies, which contributes to the limited discussion on the topic and the proliferation of many misconceptions about the effects of masturbation, implying disinformation [71].

Most of the participants in the reviewed studies are heterosexuals, which agrees with what has been generally observed in the sexuality research area [72]. This scenario reveals that sexual minorities are less represented. In this regard, the importance of integrating groups affected by social stigma in research is highlighted [73].

Solitary masturbation was assessed mostly with one ad hoc item that identified the presence/absence of masturbation or its frequency. Masturbation frequency has been stressed as a relevant measure for investigating masturbation [26,74]. This relevant parameter is related to significant indicators of sexual well-being, highlighting its relevance to sexual functioning. In women, the frequency of masturbation is positively related to orgasm pleasure [75] and to the greater facility of reaching orgasm in older women [74]. In men, more frequent masturbation is associated with more difficulty reaching an orgasm [74] and more symptoms of retarded ejaculation [76]. Therefore, this parameter has contributed to expanding scientific knowledge about masturbation and delving deeper into the study of this behavior [2]. However, we should bear in mind the diversity of time ranges and the responses employed to measure this parameter when comparing and generalizing the results reported in the present systematic review.

Sexual satisfaction was often assessed with ad hoc items about the level of experienced satisfaction. This matter has been criticized by Sánchez-Fuentes et al. [19]. Using a single item can present measurement stability problems [77], and it may generate sources of error when simplifying the evaluated construct [78]. Four works employed items taken from standardized scales, which does not guarantee suitable psychometric properties for the original instrument. Only 27% of the studies evaluated sexual satisfaction using standardized scales, which ensure that acceptable and reliable measures are obtained [79]. Of these scales, the Female Sexual Quotient [43], the Male Sexual Quotient [56], and the Multidimensional Sexuality Questionnaire (MSQ) [45] appeared. We stress the Global Measure of Sexual Satisfaction [54], used in two studies. It is a measure included in the Interpersonal Exchange Model of Sexual Satisfaction Questionnaire (IEMSSQ) [80] that derives from the Interpersonal Exchange Model of Sexual Satisfaction (IEMSS) [23], a theoretical consolidated model of sexual satisfaction [67] that has been validated in Spain [81,82], Canada [23], and the United States [83]. Considering the complexity of the conceptualization of sexual satisfaction and the diverse ways of assessing it [84], it is highly relevant to integrate its definition to compare and delve into the study of this sexual functioning dimension [85].

In relation to the obtained findings about the relation between solitary masturbation and sexual satisfaction, the studies that jointly considered men and women pointed out a negative relation between solitary masturbation and sexual satisfaction [55,57,62,68] or no relation [57,66]. Despite some studies including gender as a covariable (e.g., [57,62]), the results must be cautiously considered given the known differences between men and women in the various parameters associated with masturbation [26,74,86,87,88,89,90].

The findings in those studies that examined the relation between solitary masturbation and sexual satisfaction in men and women separately are more interesting. Most of the studies (71.4%) that have dealt with this association in men reported a negative relation between solitary masturbation and satisfaction, as opposed to 21.4% of them that did not find a significant relation and the 7.2% that reported a positive association. Thus, a negative relation was observed mostly for men, which contrasts with the evidence showing that masturbation is a positive indicator of sexual health [1] and practicing masturbation is related to different beneficial health aspects (e.g., [91,92,93]). One of the main hypotheses that could explain this finding in men stems from the compensatory model of masturbation [8,15]. This model proposes that people resort to this behavior as a substitute for sexual dissatisfaction. Previous evidence reveals that the compensatory pattern of masturbation might be more characteristic of men than women [9,12,13,14]. To support this hypothesis, more men compared to women have reported having less desire to masturbate [94] and show a more negative attitude toward masturbation at older ages [74]. This stresses the importance of considering the negative attitude toward masturbation (see [95]) when studying this behavior to understand its implication in the sexual satisfaction experience. This finding could also be interpreted in line with the hypothesis put forward by Rowland et al. [64]. According to their hypothesis, people who masturbate may exhibit a strong auto-erotic orientation, which could make this behavior more gratifying than sexual relationships. This proposal is coherent with evidence showing that men report more solitary sexual desire than women [26,74,86], they report a higher masturbation frequency (e.g., [74,88]), and among the various reasons for practicing this behavior, sexual pleasure stands out [96]. So it is proposed that future studies which examine the relation between solitary masturbation and satisfaction should include the reasons why masturbation is practiced as a mediator variable.

The studies performed with women reflect, to a greater extent, the heterogeneity of the obtained results: 40% found no relation between solitary masturbation and sexual satisfaction, 33.3% found a negative association, and 26.7% pointed out a positive relation between both variables. This greater heterogeneity of the results obtained for women might have something to do with their sexuality compared to that of men, which is generally determined by a larger number of variables [26,97,98,99], as specifically noted for sexual satisfaction [100]. One third of the studies performed with women found a negative association between solitary masturbation and sexual satisfaction. This reveals that sexual dissatisfaction could also be a reason for them to practice masturbation [93]. Masturbating could be an indicator of feeling comfortable about one’s body and sexuality, which could raise awareness about dissatisfaction or reduce the likelihood of someone exaggerating their sexual satisfaction during sexual relationships [40]. The percentage of the studies that report a positive association between solitary masturbation and sexual satisfaction was higher in women (26.7%) than in men (7.1%). In recent decades, inhibition about female sexuality may have lowered [11,15], which would reflect the empowerment role of masturbation noted in women [101,102].

The inconsistency encountered in the obtained results could be partly due to the diversity of the employed measures, and very few of the research works assessed sexual satisfaction with instruments based on robust theoretical models that have demonstrated their invariance in the population of interest. As previously mentioned, the cultural diversity in accepting and practicing masturbation could also be a source for the variation in the results [3], as could considering neither a negative attitude toward masturbation nor the reasons for masturbating to be covariables. Not all the studies contemplated interpersonal-type variables, such as satisfaction with one’s relationship, which has been associated with both practicing masturbation [94] and sexual satisfaction [100]. Other covariables that should be considered are age, given that this behavior evolves with generational advancement [10,11,14,69], having a partner because of its association with masturbation practice [9,10], and sexual satisfaction [59]. In the exploration of the distinction between being single or in a relationship, it has been observed that in the two studies focusing exclusively on single individuals, no significant association between masturbation and sexual satisfaction was found [40,57], while in the studies that considered exclusively samples of couples, they found a positive (e.g., [15]), negative (e.g., [57]), or no relation (e.g., [66]). These findings should be approached with caution due to the diversity of terminology employed (i.e., partner, sex partner, couple, in a relationship) and the limited evidence found in single people. The importance of further study of the relation between masturbation and sexual satisfaction in single individuals is highlighted [57].

Finally, it is worth mentioning that the results must be cautiously considered because the experimental design type of the reviewed studies does not allow case–effect relations to be established. To interpret the findings of our systematic review, it is necessary to bear in mind that the reviewed studies were original scientific articles written only in Spanish and English. Thus, this systematic review did not consider other languages, types of investigations (e.g., narrative and qualitative), or other reviews. As mentioned above, the diverse criteria for masturbation frequency (e.g., the past 30 days or 6 months), the different instruments used to assess sexual satisfaction, and the sample used (mostly heterosexuals) could influence the generalizability of the results.

## 5. Conclusions

Our systematic review evidences the relation between solitary masturbation and sexual satisfaction. Although its findings in favor of a negative association are present, considering sexual differences is absolutely necessary. Thus, a more consistent pattern of negative relations is found in men, which supports the compensatory role of masturbation. Conversely, the results for women are more heterogeneous, and there are more pieces of evidence for a positive relation than for men. This finding suggests that solitary masturbation for women could be an indicator that is more related to sexual health, which would support the complementary role between both behaviors (solitary masturbation and sexual relationships). It is necessary to continue research to examine in more depth the association between masturbation and sexual satisfaction, considering partnered masturbation. In future studies, given the relevance of masturbation to sexual satisfaction, it could also be interesting to examine how different patterns of sexual activity (including solitary masturbation and sexual relationships) are associated with sexual satisfaction in a romantic relationship. It would also be relevant to use a validated theoretical model of sexual satisfaction that would also include solitary masturbation frequency and other important parameters like age of masturbation onset, reasons for masturbating, and specific measures that characterize the subjective orgasm experience achieved by masturbation or taking a negative attitude toward this behavior.

## Figures and Tables

**Figure 1 healthcare-12-00235-f001:**
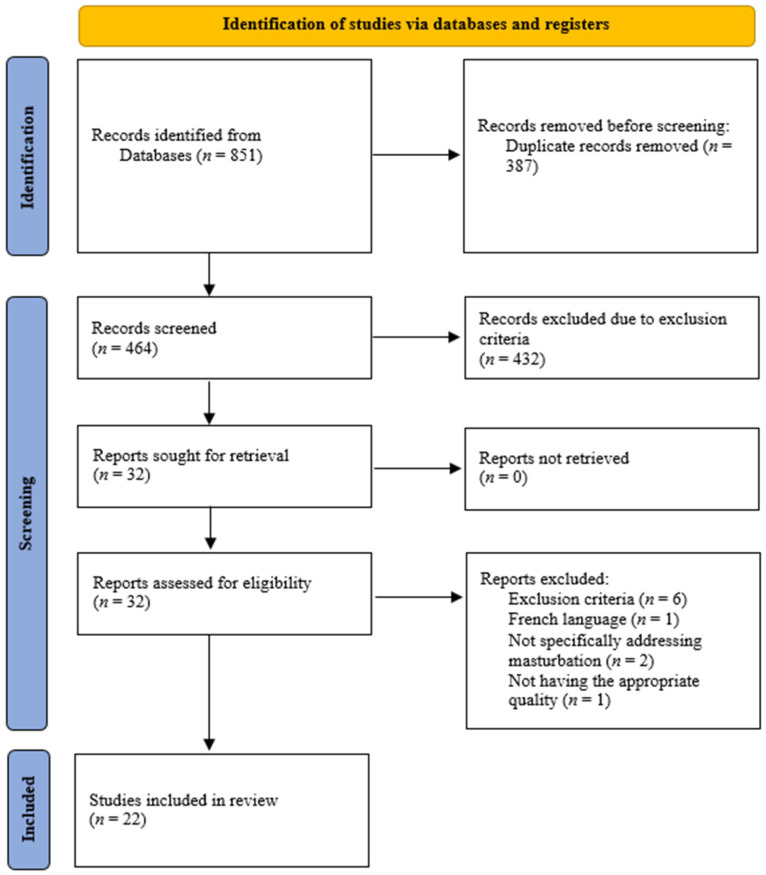
Flow diagram for the systematic review of searches of databases.

**Table 1 healthcare-12-00235-t001:** Summary of study reviews about the relationship between solitary masturbation and sexual satisfaction.

Authors	Country	Participants	Orientation	Instrument to Assess Masturbation	Instrument to Assess Sexual Satisfaction	Results
Bridges et al. [36]	United States	2632 women with a partner aged from 19 or younger to 70 or older.	Not specified.	Ad hoc item to ask about the number of times the participant has masturbated in the last month.	Four single-item ad hoc questions rated on a Likert scale of 1 (strongly disagree) to 7 (strongly agree) to assess four dimensions of sexual satisfaction: (a) “I have been satisfied with quality of genital stimulation and/or intercourse I’ve had with a partner” (stimulation/intercourse), (b) “I have been satisfied with the quality of sexual/erotic contact I’ve had with a partner that did not include or lead to sustained genital stimulation or intercourse” (sexual/erotic contact), (c) “I have been satisfied with my sex life in the last three months” (3 months), and (d)“On the whole, I have been satisfied with my sexual life” (overall satisfaction).	The frequency of masturbation is not associated with satisfaction with sexual/erotic contact, satisfaction in the last three months, or overall satisfaction (lifetime sexual satisfaction). Masturbation in the last 30 days is negatively related to satisfaction with stimulation/intercourse or genital stimulation. Covariates in regression models include family affection, partner initiation, and communication.
Brody and Costa [37]	Sweden	2810 (1255 men and 1129 women) with or without a partner, from 18 to 74 years old (*M* men = 40.9; *M* women = 40.8).	Not specified.	Ad hoc item para frequency of masturbation in the past 30 days.	Satisfaction scale comes from different versions of the Life Satisfaction Scale: LiSat-8 [38] and LiSat-11 [39]. They assess their satisfaction with their sex life on a six-point Likert-type scale, anchored with 1 = very unsatisfying and 6 = very satisfying.	The frequency of masturbation was inversely related to nearly all indices of sexual satisfaction in both men and women, with a small to medium effect size. Covariates: age, penile–vaginal intercourse, anal sex, and oral sex.
Darling and Davidson [40]	-	202 single women (*M* age = 21.6).	-	Ad hoc item: Have ever engaged in masturbation?	Ad hoc item using a Likert-scaled response.	In both sexually active and inactive women, masturbation is not associated with sexual satisfaction (psychological and physiological). No covariates.
Das [15]	United States	Subsample with a stable partner of a larger sample of 3116 individuals (1347 men and 1769 women) aged 18 to 60 years old.	Not specified.	Ad hoc item. Frequency of masturbation was assessed: “On average, in the past 12 months how often did you masturbate?” A dummy variable was used to recode the responses: any (1) or no (0) masturbation.	Ad hoc item. Dummy was included for emotional and/or physical satisfaction in sex (with the partner): Physical yes; emotional yes (reference group); Physical yes; emotional no (indicating the participant was physically but not emotionally satisfied); Physical no; emotional yes; Physical no; emotional no.	Men with a stable partner who engage in sexual relationships, as well as those who find them physically but not emotionally satisfying and those who are dissatisfied both physically and emotionally report less masturbation than those who are emotionally and physically satisfied. Women with a stable partner who have had sex in the last year, and women who find sex physically but not emotionally satisfying, report less masturbation than those who are emotionally and physically satisfied. Covariate: age.
DeLamater and Moorman [41]	United States	1384 people (745 women and 639 men) aged 45 and older. A subsample with a partner (447 women and 505 men)	99% of the members of the sample were in heterosexual relationships.	Ad hoc item to assess the frequency of masturbation. “During the past 6 months, how often, on average, have you engaged in the following sexual activities?” The behaviors included masturbation, and the response options for all items were 1 = not at all, 2 = less than once a month, 3 = once or twice a month, 4 = about once a week, 5 = more than once a week, and 6 = daily.	Two ad hoc items to assess the emotional and physical facets of sexual satisfaction. “In the past 6 months, how emotionally satisfying was your relationship with your partner?” and “In the past 6 months, how physically pleasurable was your relationship with your partner?” The options were 1 (not at all), 2 (slightly), 3 (moderately), 4 (very), and 5 (extremely).	Men who are less physically satisfied with their partners masturbate more often than men who are more physically satisfied with their partners. This is not observed in women. Covariates in the regression model: age and biopsychosocial variables.
De Lima et al. [42]	Brazil	2408 women ranging from 18 to 69 years old (*M*age = 27.78) with or without a relationship.	All options on the Kinsey scale were considered.	Ad hoc. Masturbation frequency was assessed with an item with 9 options: less than once a year, once a year, once every 6 months, once every 3 months, once or twice a month, once a week, 2 to 3 times a week, once a day, and more than once a day.	The Female Sexual Quotient instrument designed and validated in a previous project [43]. It contains 10 questions covering different areas of women’s sexual activity. The options for the 10 items are on a Likert-type scale of 5 points. The scores divide sexual performance into five categories: highly dissatisfied (0–20 points), dissatisfied (22–40 points), average (42–60 points), partially satisfied (62–80 points), and highly satisfied (82–100 points).	There is a positive correlation between masturbation frequency and sexual satisfaction.
Favez and Tissot [44]	Switzerland	244 men and 246 women aged 25–45 years old (*M* men = 36.3; *M* women = 36.3) in a committed relationship.	Not specified.	Ad hoc item: “How often do you masturbate?” Responses employed a 9-point rating scale from 1 (more than once a day) to 9 (never).	The French version of the Multidimensional Sexuality Questionnaire [45]	The frequency of solitary sex was negatively correlated with sexual satisfaction. Structural equation modeling: attachment, representation of sexuality, frequency of sexual activities and desire, and sexual satisfaction. Covariates: age, duration of the relationship, and marital satisfaction.
Fischer [46]	Norway	4148 people (2181 men, 1967 women) from 30 to +60 years old (*M* men = 48.4; *M* women = 44.4). A subsample of single (507 men, 568 women) and partnered adults (1668 men, 1395 women).	Heterosexual (87.9%), homosexual/lesbian (4.3%), bisexual/pansexual (6.5%), asexual/other (1.3%)	Ad hoc item. Masturbation frequency was assessed by a one-item indicator [47]: “How often did you masturbate in the past month?” Responses ranged from 1 = no times to 7 = more than once a day.	Ad hoc item. “All things considered, how satisfied are you with your sexual life?” with response options ranging from 1 = very dissatisfied to 5 = very satisfied.	In single men, there was no relationship between masturbation and sexual satisfaction, while in single women, a positive relationship was observed. In both men and women with a partner, there was no relationship between the frequency of masturbation and sexual satisfaction. Regression models included sociodemographic factors (age, education, self-estimated health), sexual behavioral factors (intercourse and masturbation frequency), evaluative factors (satisfaction with the relationship, contentment with sexual frequency, body image, genital image, level of sexual distress), and relationship factors (relationship duration, sexual avoidance, inclusion of the other in the self).
Fischer et al. [12]	Norway, Denmark, Belgium, and Portugal	3814 people (1875 men and 1939 women) with or without a relationship, between 60 and 75 years (*M*age 67 years).	Not specified.	Ad hoc. One-item indicator previously used to measure reported masturbation frequency (ELSA) [47]. “How often did you masturbate in the past month?” Response alternatives were 1 = none, 2 = once in the past month, 3 = 2 or 3 times in the past month, 4 = once a week, 5 = 2 or 3 times a week, 6 = once a day, and 7 = more than once a day.	Ad hoc item. “How satisfied are you with the current level of sexual activity in your life, in a general way?” Responses, which ranged from 1 = very satisfied to 5 = very dissatisfied, were reverse-recoded, so that higher scores reflected higher sexual satisfaction.	In men, greater satisfaction predicts lower levels of masturbation across all four countries (Portugal, Denmark, Norway, and Belgium). In women, being more satisfied with one’s level of sexual activity is negatively related to masturbation across all four countries (Denmark, Belgium, and Norway). Regression models included sociodemographic factors (age, education, religiosity, and relationship status), health factors, sexual behavior, and satisfaction, as well as attitudes toward sexuality.
Klapilová et al. [48]	Czech Republic	86 long-term cohabiting couples. *M*age from 20 to 40 years old (*M* men = 26.5; *M* women = 27.6).	Heterosexual.	Ad hoc item. The frequency of masturbation was assessed using a scale that ranged from 0 = never or once a year to 4 = once or more times per day.	Ad hoc item. Sexual satisfaction was rated on a seven-point Likert-type scale (1 = not at all satisfied; 7 = absolutely satisfied).	No relationship is observed between the frequency of masturbation and sexual satisfaction, neither in correlations nor in regression models, while controlling for the frequency of other variables under consideration.
Kvalem et al. [49]	Norway	2587 people (1105 men, 1482 women) in a relationship. Representative sample from 14 to 33 years old.	Not specified.	Ad hoc item. Two questions about the frequency of masturbation during the last six months: “Masturbation (of yourself)”. Response options ranged from (1) never to (6) once a day or more.	Ad hoc item. “During the last six months, how satisfied have you been with…: “Your capacity to let go during sex”; “Your capacity of feeling sexual desire”; and “The quality of your sex life.” The response categories were (1) clearly unsatisfactory, (2) slightly unsatisfactory, (3) satisfactory, (4) very good, (5) could not have been better, and (6) have not had a sex partner.	In men, a negative relationship is observed between masturbation and sexual satisfaction; in women, a positive relationship is observed between masturbation and sexual satisfaction. Age is controlled. Covariates in the regression model: body satisfaction, Body Mass Index, relationship status, intercourse activity, and mental health.
Lee et al. [47]	United Kingdom	6201 (2745 men, 3456 women) aged 50 years and older (*M* men = 66.9; *M* women = 66.8). A subsample in a partnership (2009 men, 2053 women).	Heterosexual	Question extracted from the ELSA Sexual Relationships and Activities Questionnaire (SRA-Q) included in the ELSA project. An instrument with items from the Natsal-SF [50], the European Male Ageing Study Sexual Function Questionnaire [51], and the NSHAP Project [52]. The item was how often did you masturbate?	Ad hoc item. In the context of partnership satisfaction, “How satisfied have you been with your overall sex life?”. Responses ranged on a 5-point scale from very satisfied to very dissatisfied. Those who answered moderately dissatisfied or very dissatisfied were classified as dissatisfied.	In men, a positive relationship is observed between the frequency of masturbation and being dissatisfied with their overall sex life. In women, no significant relationship is found. Adjusted for age and self-rated health.
Miller et al. [53]	Australia	661 men (two samples of 326 and 335) (*M*age = 27.63 and 46.76, respectively). A subsample of partnered men with sexual relationships (Study 1: 156; Study 2: 320).	Heterosexual	Ad hoc item. Frequency of masturbation over the past 6 months. To respond, an 8-point scale was used (where 1 = less than monthly, 2 = monthly, 3 = fortnightly, 4 = 1–2 times per week, 5 = 3–4 times per week, 6 = 5–6 times per week, 7 = daily, and 8 = more than once a day).	Sexual satisfaction was measured using the Global Measure of Sexual Satisfaction and the Global Measure of Relationship Satisfaction [54]. Participants rated their sexual relationship and overall relationship across three 7-point bipolar scales: good–bad, satisfying–unsatisfying, and valuable–worthless. An overall sexual satisfaction score was calculated.	In Study 1 and Study 2, masturbation frequency is significantly negatively associated with sexual satisfaction.
Neto et al. [55]	Brazil	1314 people (386 men, 928 women) with a mean age of 37.6 years old (*M* = 37.6) with a partner.	Heterosexual and homosexual.	Ad hoc item. Before and after the quarantine, an ordinal multiple-choice question (<1, 1–2, 3–5, >5/week) graded the masturbatory sexual frequency.	The Female Sexual Quotient (FSQ) [43] and the Male Sexual Quotient (MSQ) [56] were used. Both instruments were developed in Brazilian Portuguese. Questions express the satisfaction level, contemplating the sexual response cycle phases. Responses ranged from 0 = never to 5 = always.	A higher frequency of masturbation is associated with poorer sexual satisfaction in both men and women. Covariates in the regression model: lack of nightlife (clubs, bars, restaurants), lower libido, isolation from partner, working at a central institute, higher sexual frequency, and sexually active.
Park and MacDonald [57]	-	Study 1: 489 participants (264 men, 223 women, 2 unidentified) who were 27.81 years old, with an average age ranging from 18 to 70. Half of the participants were in a relationship. Study 2: 286 single people (150 men, 136 women), M age = 37.72, from 19 to 79 years old, including 463 partnered individuals (257 men, 203 women, and 2 others) from 19 to 79, *M*age = 39.	Not specified.	Ad hoc item: “Please rate how often you DO or GET each of the listed sexual activities: sexually touching myself (e.g., masturbation).” Responses ranged from 1 = not at all to 7 = a lot.	Four questions from the Satisfaction with Sex Life Scale—Revised [58] and from [45]: In most ways, my sexual life is close to my ideal. The conditions of my sexual life are excellent. I am satisfied with my sexual life. My sexual life meets my expectations. Responses ranged from 1 = Not at all to 7 = extremely.	Study 1: a significant negative association between masturbation frequency and sexual satisfaction was observed only in individuals with a partner (vs. singles). Study 2: neither in the correlations nor in the regression model was a significant relationship observed between masturbation and sexual satisfaction. Covariates in regression models: gender, age, solitary desire, partnered desire, partnered activity, and interactions.
Pedersen and Blekesaune [59]	Norway	1303 men, 1508 women (age 20–26). In a subsample of 2101 that had a sex partner.	Not specified.	Ad hoc item. Masturbation frequency was asked. Responses were less than monthly, once per month, to 2–6 times a week, daily, or more often.	Four questions about sexual functioning and sexual relationships are based on the Derogatis Interview for Sexual Functioning (DISF-SR) [60] and the Changes in Sexual Functioning Questionnaire (CSF) [61]. “During the last six months, how satisfied have you been with…:” Your own capability to give yourself when you have sex. Your own capability to experience sexual lust. The quality of your sex life. The total relationship with current or last sex partner. Responses to sexual satisfaction items on a 6-point scale ranged from could not have been better to clearly unsatisfactory with an answer option for having not had any sex partner.	Masturbation frequency is negatively associated with sexual satisfaction in men, but not in women. Covariates in the regression model: age, partner status, relationship duration, social support, masculinity/femininity, depression/anxiety, intercourse debut age, kissing/hugging, intercourse/oral sex, extra-dyadic relationship, and lifetime sex partners.
Phuah et al. [62]	Malasya	621 participants (39.5% men, 60.5% women) aged 18 to 30 (*M* = 22.1). Participants without a sex partner were excluded from analysis.	Not specified.	Ad hoc item to rate their frequency of masturbation using a scale from 1 = never to 8 = multiple times a day.	One item from the Female Sexual Function Inventory (FSFI) [63]: “Over the past 4 weeks, how satisfied have you been with your overall sexual life?” referring to the past four weeks. Responses ranged from 1 = very dissatisfied to 5 = very satisfied.	Masturbation frequency was negatively associated with sexual satisfaction. Covariates in the regression model: age, SES, gender, frequency of partnered sex, availability of partner, and religiosity.
Rowland et al. [64]	United States, other English-speaking countries (e.g., Canada, England) and Hungary.	Subsample of 3343 participants from a sample of 3586 men who had had a sexual partner or were having sex with their partner, aged 18 to 85 (*M* = 40.8).	Not specified.	Ad hoc item asking about frequency of masturbation ranging from 0 = never to 10 = more than 4×/day.	Ad hoc item to assess sexual satisfaction: “how satisfied are you with the sexual aspects of your relationship”. Responses ranged from 1 = not satisfied at all to 5 = very satisfied.	Higher masturbation frequency was associated with lower sexual satisfaction. Covariates in the regression model: age, medical issue, anxiety, frequency of pornography use, sexual interest, delayed ejaculation, and erectile dysfunction.
Tao and Brody [65]	China	158 participants (84 men, 74 women) aged 24 years or older.	Not specified.	Ad hoc item. Days in the past month engaged in masturbation, and days in past month orgasm from activity of masturbation.	Sexual satisfaction was measured in two ways: The full sexual satisfaction scale of the Multidimensional Sexuality Questionnaire (MSQ) [45]. A single item from the scale “I am very satisfied with the sexual aspects of my life”, similar to the single item from the Life Satisfaction Scale LiSat-11 [39] used in the Swedish study [37]. Both are rated on a five-point scale of agreement to disagreement.	No predictive capacity of masturbation is observed to explain the two measures of sexual satisfaction, both separately and together, in both men and women.
Vaillancourt-Morel et al. [66]	Canada	211 couples (247 women, 174 men, and 1 intersex who identified as a man) aged from 18 to 70 years.	72 same-sex couples (34.1%; 54 women–women and 18 men–men) and 139 mixed-sex couples. Heterosexual (57.1%; *n* = 241); bisexual (11.4%; *n* = 48); gay/lesbian (16.8%, *n* = 71); queer (8.5%, *n* = 36); pansexual (4.0%, *n* = 17); and (2.1%, *n* = 9) as “other”, including asexual or uncertain.	Ad hoc item. Participants were asked whether they had sexual activity alone that included masturbation within the last 24 h or since they last completed a diary. This item was coded as 0 = no masturbation today and 1 = masturbation today.	Global Measure of Sexual Satisfaction [67] was used to evaluate participants’ general global satisfaction.	There is no observed relationship between masturbation and sexual satisfaction for ‘actor’ and ‘partner’ separately. Neither self-masturbation nor partner’s masturbation had the capacity to explain sexual satisfaction. The use of pornography (yes/no) is controlled for.
Velten and Margraf [16]	Germany	964 couples (1928 people) from 18 to 90 years old (*M* = 51.28).	98% heterosexual couples, 0.9% male–male, 0.5% female–female.	Ad hoc item. The frequency of masturbation was assessed on a 6-point scale: never, less than once a month, once to three times a month, once to twice a week, three to four times a week, and five times a week and more.	Ad hoc item. A single item assesses the degree to which participants were satisfied with their sexual lives. It was answered on a scale ranging from 0 to 100, with lower scores indicating lower satisfaction.	In both men and women, an actor effect of masturbation frequency was observed to explain sexual satisfaction negatively. There was no partner effect of masturbation on sexual satisfaction. Covariates in the APIM model: sexual function, sexual distress, desire discrepancy, sexual initiation, sexual communication, sociosexual orientation, age, relationship duration, and household income.
Wang et al. [68]	United States	1670 from general population and 166 athletes (47.4%/53.6%).	Not specified.	Ad hoc item. “During the past 6 months, how often, on average, have you engaged in the following sexual activities?” Responses ranged from 1 = not at all to 6 = daily.	Ad hoc item: “How satisfied are you with you sex life?” The response categories were extremely dissatisfied, somewhat dissatisfied, neither satisfied nor dissatisfied, somewhat satisfied, and extremely satisfied.	The frequency of solitary sexual activity was negatively associated with sexual satisfaction. Covariates in the regression model: negative attitudes toward sex, partner-involved sexual activities, self-sexual activity and sexual desire, quality of life, height, orgasm frequency, positive attitudes toward sex, sexual desire, health, exercise, quality of life, and self-stimulation.

## Data Availability

Data are contained within the article.

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
