# Peer review of "Relationship between Solitary Masturbation and Sexual Satisfaction: A Systematic Review"

_healthcare, 2024, doi:10.3390/healthcare12020235_

Round 1

Reviewer 1 Report

Comments and Suggestions for Authors

The authors do a good job in describing their study, conducting it to the PRISMA standards, and report their findings, discussing its implications to what is expected. Nevertheless, there are several minor aspects that need revision, yet none of them are of major concern. I leave a detailed list of these concerns below.

I do hereby submit the manuscript to be accepted after minor revisions. Congratulations.

Title and Abstract

I believe the work is about “solitary” masturbation, and not just masturbation. Researchers should change their title to reflect that.

Line 9 = Authors may consider changing “sexual health” for “sexual wellbeing”. The former is a bit more general and more medical oriented than the latter, which is closer to sexual satisfaction.

Line 12 = I believe researchers could indicate how many initial articles they got from the search, since it is, at this point, a bit incredible to see they review only 22 articles for such topic.

It would not hurt to clarify for the reader in the abstract if “masturbation” was evaluated in single people or those in a relationship, or both.

Introduction

Line 28 = whereas indeed masturbation can be solitary, it can also be with other people. None is more correct than the other.

The end of Line 30 is weird. Revise

Line 48 = I do not believe satisfaction is part of the cycle, but a consequence of it. Even though, to the knowledge of this reviewer, there is only one model that considers “sexual satisfaction” as part of the cycle, no other does so. Therefore, if the authors wish to keep it as it is, they should specify for which model. Also, I believe the authors are capable to paraphrase the definition.

Whereas it is not part of the study, authors need to acknowledge that there is also partner masturbation in their introduction.

Finally, I believe authors need to specify they aim to assess binary gender differences in their study.

Method

Line 73 = do the authors mean examined “directly” the relation?

Line 111 = authors need a reference for this Kappa criterion.

Results

It is rather incredible the searched yielded only 851 records. Authors should include a list of all titles found as a supplementary table for transparency.

Line 135 = it is uncertain how authors got from 464 to 32 records.

There is no report on sexual orientation diversities.

Discussion

Line 226 = I believe authors need to further explain, at least a bit, how taboo implies disinformation. The sentence is a bit too short, leading the reader to fill/assume the blank.

Line 224 = The analysis on solitary masturbation frequency is just too short. Authors need to delve into it for it makes a significant part of the picture in this phenomenon.

Line 251 = I do not believe “therefore” is an appropriate way to begin a paragraph. It denotes a paragraph has not been properly wrapped up or a new one not properly began.

The content of the paragraph between lines 251 and 275 is of great value. Good job.

Line 276 and further = Men’ sexuality is as complex as women’s. An argument like that with not age well. Much like women research have been underrepresented for years in sex research, men have been stereotyped of being penis- and ejaculation-oriented creatures lacking complexity in their response and wellbeing. I suggest authors to reconsider their statement.

Line 294 = lack of measurement invariance of the scales is a likely candidate, too.

There is no discussion on the distinction of solitary masturbation in the context of being single or in a relationship.

I believe authors need to straighten their manuscript to refer always to solitary masturbation and not just masturbation.

Limitations and future studies

Line 316 = how should the research to examine in more depth the association?

Authors only analyzed heterosexual men and women gender differences, without even acknowledging other genders or sexual orientation. This is an important limitation to acknowledge. Similarly, partner masturbation implication could be suggested as part of future studies.

Author Response

The authors do a good job in describing their study, conducting it to the PRISMA standards, and report their findings, discussing its implications to what is expected. Nevertheless, there are several minor aspects that need revision, yet none of them are of major concern. I leave a detailed list of these concerns below.

I do hereby submit the manuscript to be accepted after minor revisions. Congratulations.

>>Response. We appreciate the comments and suggestions provided by the reviewer, which have allowed us to enhance and delve deeper into the manuscript. Below, we are addressing each of their contributions one by one.

Title and Abstract

I believe the work is about “solitary” masturbation, and not just masturbation. Researchers should change their title to reflect that.

>>Response. Thank you for your suggestion. Including "solitary" helps to further clarify the aim of this systematic review. It has been added to the title and abstract.

Line 9 = Authors may consider changing “sexual health” for “sexual wellbeing”. The former is a bit more general and more medical oriented than the latter, which is closer to sexual satisfaction.

>>Response. Thank you for your recommendation. The change has been made to "sexual wellbeing" with the purpose of including this term more associated with sexual satisfaction.

Line 12 = I believe researchers could indicate how many initial articles they got from the search, since it is, at this point, a bit incredible to see they review only 22 articles for such topic.

>>Response. We appreciate your suggestion. The number of initial articles has been included in lines 12-13.

It would not hurt to clarify for the reader in the abstract if “masturbation” was evaluated in single people or those in a relationship, or both.

>>Response. The objective of this systematic review focuses on the behavior of solitary masturbation in both partnered and single individuals, without being an exclusion criterion. This information is now incorporated into the Abstract, and Table 1 also presents these details.

Introduction

Line 28 = whereas indeed masturbation can be solitary, it can also be with other people. None is more correct than the other.

>>Response. We agree with your comment; solitary masturbation is not more correct than partnered masturbation. This question is clarified in line 28. Thank you for emphasizing this matter. In the case of this review, the definition in line 29 emphasizes the specific goal of studying solitary masturbation activity (in individuals who may be single or in a relationship).

The end of Line 30 is weird. Revise.

>>Response. Thank you for your guidance; it has been clarified in the manuscript.

Line 48 = I do not believe satisfaction is part of the cycle, but a consequence of it. Even though, to the knowledge of this reviewer, there is only one model that considers “sexual satisfaction” as part of the cycle, no other does so. Therefore, if the authors wish to keep it as it is, they should specify for which model. Also, I believe the authors are capable to paraphrase the definition.

>>Response. Once again, we appreciate your comment, and it has been clarified that we rely on the consideration of the Basson’s model, as cited in the manuscript (line 51). The direct quotation is intended to preserve the original meaning of the definition.

Whereas it is not part of the study, authors need to acknowledge that there is also partner masturbation in their introduction.

>>Response. Following your recommendation, masturbation with a partner has been mentioned in line 28.

Finally, I believe authors need to specify they aim to assess binary gender differences in their study.

>>Response. Thank you very much for your suggestion because indeed the description of the relationship between solitary masturbation and sexual satisfaction in men and women has been addressed. The objective has been specified in lines 64-65.

 Method

Line 73 = do the authors mean examined “directly” the relation?

>>Response. The aim of the review was to examine studies that analyze the relation between solitary masturbation and sexual satisfaction either directly or taking into account possible involved covariates. This information is now provided in lines 76-77.

Line 111 = authors need a reference for this Kappa criterion.

>>Response. Following your recommendation, a reference for the Kappa criterion has been included. Pendiente de incluir en el manuscrito en función de si modifican más referencias.

Monteiro, A., Vázquez, M.J., Seijo, D., y Arce, R. (2018). ¿Son los criterios de realidad válidos para clasificar y discernir entre memorias de hechos auto-experimentados y de eventos vistos en vídeo? [Are the reality criteria valid to classify and to discriminat between memories of self-experienced events and memories of video-observed events?]. Revista Iberoamericana de Psicología y Salud, 9,149-160. https://doi.org/10.23923/j.rips.2018.02.020

Results

It is rather incredible the searched yielded only 851 records. Authors should include a list of all titles found as a supplementary table for transparency.

>>Response. Thank you for your guidance. As Supplementary Material, a list of the search results, including the title, authors, and publication year of the studies, is attached.

Line 135 = it is uncertain how authors got from 464 to 32 records.

>>Response. This issue has been clarified in the manuscript (lines 142-143).

There is no report on sexual orientation diversities.

>>Response. Table 1 reports the sexual orientation of the sample of those articles that indicated it. In addition, lines 161-164 report the sexual orientation of the studies reviewed. 

Discussion

Line 226 = I believe authors need to further explain, at least a bit, how taboo implies disinformation. The sentence is a bit too short, leading the reader to fill/assume the blank.

>>Response. Thank you for your comment. How taboo implies disinformation has been further explained in the manuscript (lines 222-224).

Line 224 = The analysis on solitary masturbation frequency is just too short. Authors need to delve into it for it makes a significant part of the picture in this phenomenon.

>>Response. Thank you for the indication. This aspect has been expanded in the manuscript (lines 231-238).

Line 251 = I do not believe “therefore” is an appropriate way to begin a paragraph. It denotes a paragraph has not been properly wrapped up or a new one not properly began.

>>Response. Thank you for your suggestion; line 264 has been modified.

The content of the paragraph between lines 251 and 275 is of great value. Good job.

>>Response. We appreciate your review comment of this paragraph.

Line 276 and further = Men’ sexuality is as complex as women’s. An argument like that with not age well. Much like women research have been underrepresented for years in sex research, men have been stereotyped of being penis- and ejaculation-oriented creatures lacking complexity in their response and wellbeing. I suggest authors to reconsider their statement.

Response. Thank you for your guidance; it has been removed.

Line 294 = lack of measurement invariance of the scales is a likely candidate, too.

>>Response. Thank you very much for your recommendation, which provides further justification. It has been included in line 307-308.

There is no discussion on the distinction of solitary masturbation in the context of being single or in a relationship.

>>Response. Thank you for the suggestion; more information discussing this issue has been included in lines 316-324.

I believe authors need to straighten their manuscript to refer always to solitary masturbation and not just masturbation.

>>Response. Thank you for the recommendation; this matter has been reviewed.

Limitations and future studies

Line 316 = how should the research to examine in more depth the association?

>>Response. Thank you very much for the comment. Future research suggestion have been expanded in the conclusions section, in lines 344-347.

Authors only analyzed heterosexual men and women gender differences, without even acknowledging other genders or sexual orientation. This is an important limitation to acknowledge. Similarly, partner masturbation implication could be suggested as part of future studies.

>>Response. Thank you for your comment. The discussion between men and women does not exclusively refer to heterosexual orientation; all articles are included regardless of sexual orientation. We agree that the study of solitary masturbation and sexual satisfaction in a population diverse from heterosexual is limited. This issue is addressed in lines 225-228. We have also introduced the idea of expanding this study to include partnered masturbation, as mentioned in line 344.

Reviewer 2 Report

Comments and Suggestions for Authors

Dear authors,

It was a pleasure for me to review this manuscript that attempts to establish the relationship between masturbation and sexual satisfaction through a systematic review.

I would like to congratulate the authors for exploring such interesting topics as masturbation and satisfaction for people's sexual health.

With the sole purpose of improving the quality of the manuscript I would like to make some comments.

The introduction section seemed correct to me, since it places us in the context of the problem using appropriate and updated bibliographic sources.

The material and method section also seemed correct and well structured to me. However, in point 2.7 where the risk of bias of the studies was assessed, it was not entirely clear to me how the articles were selected based on risk of bias. I would imagine that the tools used to measure this risk would assign a score to each document. I think it would be interesting to explain at what score each document was eligible and below what score it was excludable.

The results were clearly explained and each of the results obtained was correctly discussed in the discussion. However, in the last paragraph of the discussion reference is made to certain limitations of the study. I think it would be appropriate to rewrite this paragraph so that it is clearly and explicitly explained what the limitations of the study were.

To conclude, I would like to say that the conclusions seemed correct to me, responding perfectly to the objective of the research.

In conclusion, I find it a very interesting manuscript for the knowledge of sexual health.

Thanks

Author Response

Dear authors,

It was a pleasure for me to review this manuscript that attempts to establish the relationship between masturbation and sexual satisfaction through a systematic review.

I would like to congratulate the authors for exploring such interesting topics as masturbation and satisfaction for people's sexual health.

With the sole purpose of improving the quality of the manuscript I would like to make some comments.

The introduction section seemed correct to me, since it places us in the context of the problem using appropriate and updated bibliographic sources.

The material and method section also seemed correct and well structured to me. However, in point 2.7 where the risk of bias of the studies was assessed, it was not entirely clear to me how the articles were selected based on risk of bias. I would imagine that the tools used to measure this risk would assign a score to each document. I think it would be interesting to explain at what score each document was eligible and below what score it was excludable.

The results were clearly explained and each of the results obtained was correctly discussed in the discussion. However, in the last paragraph of the discussion reference is made to certain limitations of the study. I think it would be appropriate to rewrite this paragraph so that it is clearly and explicitly explained what the limitations of the study were.

To conclude, I would like to say that the conclusions seemed correct to me, responding perfectly to the objective of the research.

In conclusion, I find it a very interesting manuscript for the knowledge of sexual health.

Thanks

Response.

We appreciate the reviewer's comments of this work. The suggestions provided are very useful for improving this study. We discuss them below:

The tools used in section 2.7 do not have a specific cutoff point. In this work, we reviewed the sections to ensure the quality and robustness of the results indicated in each study included in the systematic review. In point 2.7, we have expanded the criteria considered to examine the methodological quality of the articles.

Regarding the limitations, we appreciate your suggestion, and they have been modified and expanded for this systematic review.